# Enhancing Diffusion-Based Image Synthesis with Robust Classifier Guidance

**Bahjat Kawar**                                            *bahjat.kawar@cs.technion.ac.il*
*Computer Science Department*
*Technion, Israel*

**Roy Ganz**                                                *ganz@campus.technion.ac.il*
*Electrical Engineering Department*
*Technion, Israel*

**Michael Elad**                                            *elad@cs.technion.ac.il*
*Computer Science Department*
*Technion, Israel*

**Reviewed on OpenReview:** *https://openreview.net/forum?id=tEVpz2xJWX*

## Abstract

Denoising diffusion probabilistic models (DDPMs) are a recent family of generative models that achieve state-of-the-art results. In order to obtain class-conditional generation, it was suggested to guide the diffusion process by gradients from a time-dependent classifier. While the idea is theoretically sound, deep learning-based classifiers are infamously susceptible to gradient-based adversarial attacks. Therefore, while traditional classifiers may achieve good accuracy scores, their gradients are possibly unreliable and might hinder the improvement of the generation results. Recent work discovered that adversarially robust classifiers exhibit gradients that are aligned with human perception, and these could better guide a generative process towards semantically meaningful images. We utilize this observation by defining and training a time-dependent adversarially robust classifier and use it as guidance for a generative diffusion model. In experiments on the highly challenging and diverse ImageNet dataset, our scheme introduces significantly more intelligible intermediate gradients, better alignment with theoretical findings, as well as improved generation results under several evaluation metrics. Furthermore, we conduct an opinion survey whose findings indicate that human raters prefer our method's results.

## 1 Introduction

Image synthesis is one of the most fascinating capabilities that have been unveiled by deep learning. The ability to automatically generate new natural-looking images without any input was first enabled by revolutionary research on VAEs – variational auto-encoders (Kingma & Welling, 2014) and GANs – generative adversarial networks (Goodfellow et al., 2014). Both these techniques, as well as their many subsequent works (Radford et al., 2016; Arjovsky et al., 2017; Gulrajani et al., 2017; Karras et al., 2020; Van Den Oord et al., 2017; Vahdat & Kautz, 2020), involved training neural networks on a large dataset of natural images, aiming to convert simple and easily accessed random vectors into images drawn from a distribution close to the training set. Despite their impressive capabilities, the image distributions learned by these models were initially restricted to a specific class of images, ranging from low-resolution handwritten digits (Deng, 2012) to higher-resolution human faces (Karras et al., 2019). As more research and resources were invested in this field, several works (Pu et al., 2017; Brock et al., 2018; Esser et al., 2021) were able to make a leap forward, and devise more complicated models capable of synthesizing a diverse range of natural images. A commonly agreed upon challenge in this context is ImageNet (Deng et al., 2009), a dataset containing millions of natural

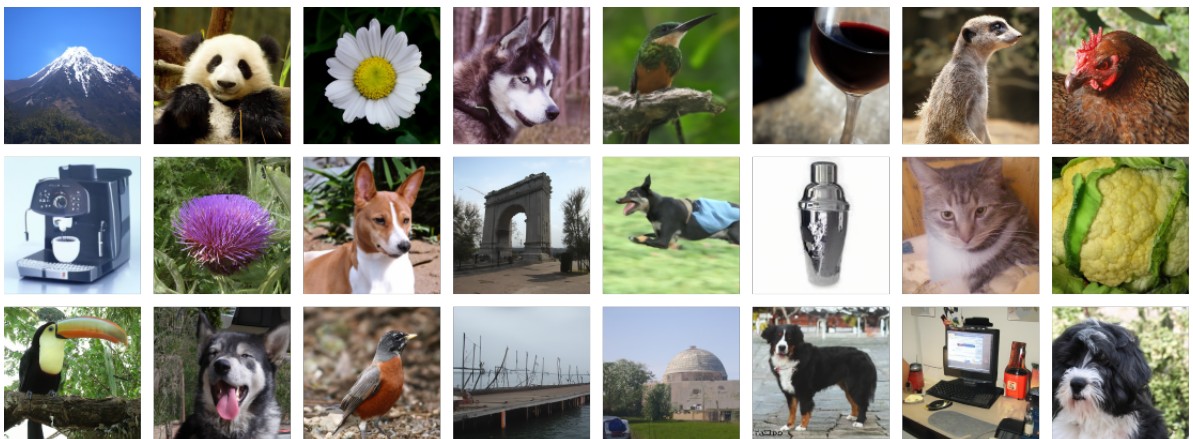

Figure 1: Images generated with our proposed method.

images, all labeled with one of 1000 image classes. For a given class label (*e.g.* "hen" or "baseball"), these class-conditional generative models can nowadays synthesize a realistic image of that class.

Recently, a different family of generative models has emerged to the forefront of image synthesis research. Denoising diffusion probabilistic models (Sohl-Dickstein et al., 2015; Ho et al., 2020), also known as score-based generative models (Song & Ermon, 2019), have achieved new state-of-the-art image generation performance (Dhariwal & Nichol, 2021; Song et al., 2021; Vahdat et al., 2021), showcasing better image fidelity and mode coverage than VAEs and GANs. These models have also excelled at several downstream tasks (Amit et al., 2021; Kawar et al., 2021b; Theis et al., 2022; Nie et al., 2022), and they also act as the powerhouse behind the unprecedented capabilities of text-to-image models (Ramesh et al., 2022; Saharia et al., 2022). Essentially, these models utilize a Gaussian denoising neural network in an iterative scheme – starting from a pure Gaussian noise image, it is continually and gradually denoised in a controlled fashion, while also being perturbed randomly, until it finally turns into a synthesized natural-looking image. To achieve class-conditional generation, the denoising neural network can accept the class label as input (Ho et al., 2022a). Additionally, the diffusion process can be guided by gradients from a classifier (Dhariwal & Nichol, 2021). This brings us naturally to discuss the next topic, of image classifiers, and their role in image synthesis.

In parallel to the progress in image synthesis research, substantial efforts were also made in the realm of image classification. Given an input image, neural networks trained for classification are able to assign it a class label from a predefined set of such labels, often achieving superhuman performance (He et al., 2016; Dosovitskiy et al., 2020). Despite their incredible effectiveness, such classifiers were found to be susceptible to small malicious perturbations known as adversarial attacks (Szegedy et al., 2014). These attacks apply a small change to an input image, almost imperceptible to the human eye, causing the network to incorrectly classify the image. Subsequently, several techniques were developed for defending against such attacks (Madry et al., 2018; Andriushchenko & Flammarion, 2020; Zhang et al., 2019; Wang et al., 2020), obtaining classifiers that are *adversarially robust*. In addition to their resistance to attacks, robust classifiers were also found to possess unexpected advantages. The gradients of a robust classifier model were found to be *perceptually aligned*, exhibiting salient features of a class interpretable by humans (Tsipras et al., 2019). This phenomenon was harnessed by a few subsequent works, enabling robust classifiers to aid in basic image generation, inpainting, and boosting existing generative models (Santurkar et al., 2019; Ganz & Elad, 2021).

In this work, we draw inspiration from these recent discoveries in image classification and incorporate their advantages into the world of diffusion-based image synthesis, which has largely been oblivious to the capabilities of robust classifiers. Dhariwal & Nichol (2021) suggested to use gradients from a (non-robust) classifier for guiding a diffusion synthesis process. We improve upon this technique by examining the validity of these gradients and suggesting a way to obtain more informative ones. We pinpoint several potential issues in the training scheme of classifiers used as guidance and observe their manifestation empirically (see Figures 3 and 4). We then propose the training of an adversarially robust, time-dependent classifier (*i.e.*, a classifier

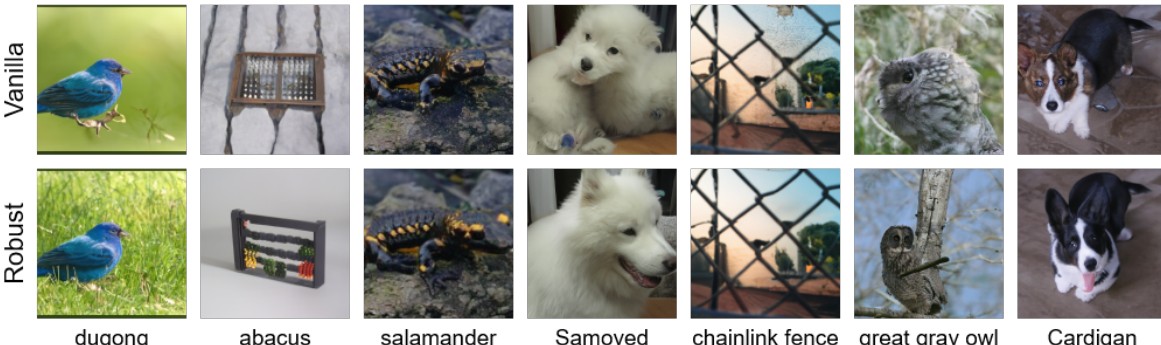

Figure 2: Images generated by guided diffusion using the same random seed and class label, with a vanilla (top) and a robust (bottom) classifier. Our robust model provides more informative gradients, leading to better synthesis quality.

that accepts the current diffusion timestep as input) suited for guiding a generation process. Our proposed training method resolves the theoretical issues raised concerning previous classifier guidance techniques. Empirically, our method attains significantly enhanced generative performance on the highly challenging ImageNet dataset (Deng et al., 2009). We evaluate the synthesis results using standard metrics, where our method outperforms the previous state-of-the-art classifier guidance technique. Furthermore, we conduct an opinion survey, where we ask human evaluators to choose their preferred result out of a pair of generated images. Each pair consists of two images generated using the same class label and the same random seed, once using the baseline classifier guidance method (Dhariwal & Nichol, 2021), and once using our proposed robust classifier guidance. Our findings show that human raters exhibit a pronounced preference towards our method's synthesis results.

To summarize, we incorporate a recently discovered capability of robust classifiers, perceptually aligned gradients, into the classifier-guided diffusion-based image synthesis scheme (Dhariwal & Nichol, 2021). We highlight several benefits of the adversarial training scheme and show how they can aid in classifier guidance for diffusion models. To that end, we train an adversarially robust time-dependent classifier on the diverse ImageNet dataset (Deng et al., 2009). We use this classifier in conjunction with a conditional diffusion model to obtain high quality image generation. The resulting technique outperforms the previous vanilla classifier guidance method on several key evaluation metrics such as FID (Heusel et al., 2017). Furthermore, we present a conducted opinion survey, which found that human evaluators show a clear preference towards our method.

## 2 Background

### 2.1 Robust Classifiers

Deep learning-based classifiers parameterized by $\phi$ aim to model the log-likelihood of a class label $y \in \{1, \ldots, C\}$ given a data instance $\mathbf{x} \in \mathbb{R}^d$, namely $\log p_\phi(y|\mathbf{x})$. Such architectures are trained to minimize the empirical risk over a given labeled training set $\{\mathbf{x}_i, y_i\}_{i=1}^N$, *e.g.*,

$$\min_\phi \frac{1}{N} \sum_{i=1}^N \mathcal{L}_{CE}(\mathbf{h}_\phi(\mathbf{x}_i), y_i), \tag{1}$$

where $N$ is the number of training examples, $\mathbf{h}_\phi(\mathbf{x}_i) = \{\log p_\phi(j|\mathbf{x}_i)\}_{j=1}^C$ is the set of these log-likelihood scores predicted by the classifier for the input $\mathbf{x}_i$, and $\mathcal{L}_{CE}$ is the well-known cross-entropy loss, defined as

$$\mathcal{L}_{CE}(\mathbf{z}, y) = -\log \frac{\exp(\mathbf{z}_y)}{\sum_{j=1}^C \exp(\mathbf{z}_j)}. \tag{2}$$

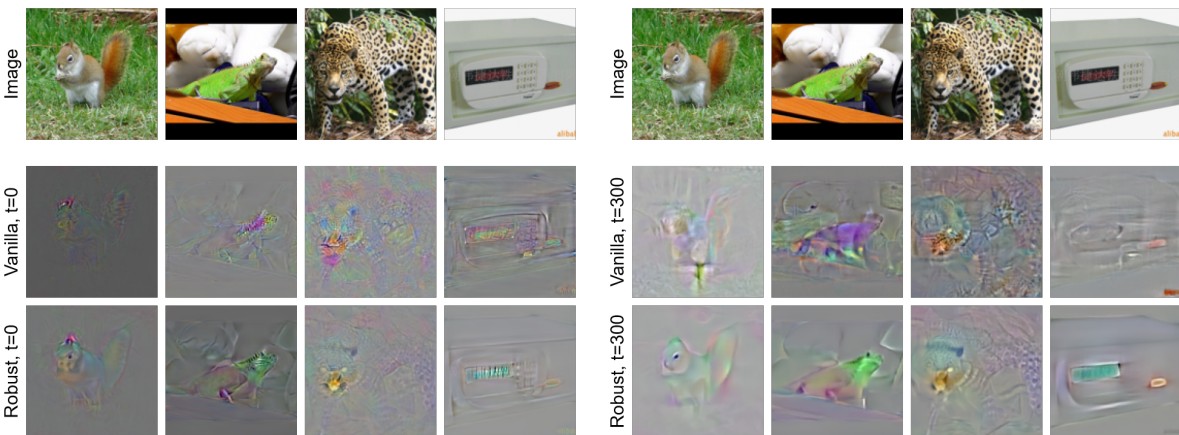

Figure 3: Gradients of images on their respective true class labels, using a vanilla classifier and our robust one at different timesteps. Gradients are min-max normalized.

Classifiers of this form have had astounding success and have led to state-of-the-art (SOTA) performance in a wide range of domains (He et al., 2016; Simonyan & Zisserman, 2014). Nevertheless, these networks are known to be highly sensitive to minor corruptions (Hosseini et al., 2017; Dodge & Karam, 2017; Geirhos et al., 2017; Temel et al., 2017; 2018; Temel & AlRegib, 2018) and small malicious perturbations, known as *adversarial attacks* (Szegedy et al., 2014; Athalye et al., 2017; Biggio et al., 2013; Carlini & Wagner, 2017; Goodfellow et al., 2015; Kurakin et al., 2017; Nguyen et al., 2014). With the introduction of such models to real-world applications, these safety issues have raised concerns and drawn substantial research attention. As a consequence, in recent years there has been an ongoing development of better attacks, followed by the development of better defenses, and so on.

While there are abundant attack and defense strategies, in this paper we focus on the Projected Gradient Descent (PGD) attack and Adversarial Training (AT) robustification method (Madry et al., 2018) that builds on it. PGD is an iterative process for obtaining adversarial examples – the attacker updates the input instance using the direction of the model's gradient w.r.t. the input, so as to maximize the classification loss. AT is an algorithm for robustifying a classifier, training it to classify maliciously perturbed images correctly. Despite its simplicity, this method was proven to be highly effective, yielding very robust models, and most modern approaches rely on it (Andriushchenko & Flammarion, 2020; Huang et al., 2020; Pang et al., 2020; Qin et al., 2019; Xie et al., 2019; Zhang et al., 2019; Wang et al., 2020).

In addition to the clear robustness advantage of adversarial defense methods, Tsipras et al. (2019) has discovered that the features captured by such robust models are more aligned with human perception. This property implies that modifying an image to maximize the probability of being assigned to a target class when estimated by a robust classifier, yields semantically meaningful features aligned with the target class. In contrast, performing the same process using non-robust classifiers leads to imperceptible and meaningless modifications. This phenomenon is termed *Perceptually Aligned Gradients* (PAG). Since its discovery, few works (Santurkar et al., 2019; Aggarwal et al., 2020) have harnessed it for various generative tasks, including synthesis refinement as a post-processing step (Ganz & Elad, 2021).

## 2.2 Diffusion Models

Denoising diffusion probabilistic models (DDPMs), also known as simply *diffusion models*, are a family of generative models that has recently been increasing in popularity (Song & Ermon, 2019; Ho et al., 2020). These methods have demonstrated unprecedented realism and mode coverage in synthesized images, achieving state-of-the-art results (Dhariwal & Nichol, 2021; Song et al., 2021; Vahdat et al., 2021) in well-known metrics such as Fréchet Inception Distance – FID (Heusel et al., 2017). In addition to image generation, these techniques have also been successful in a multitude of downstream applications such as image restoration (Kawar et al., 2021a; 2022), unpaired image-to-image translation (Sasaki et al., 2021), image

segmentation (Amit et al., 2021), image editing (Liu et al., 2021; Avrahami et al., 2022), text-to-image generation (Ramesh et al., 2022; Saharia et al., 2022), and more applications in image processing (Theis et al., 2022; Gao et al., 2022; Nie et al., 2022; Blau et al., 2022; Han et al., 2022) and beyond (Jeong et al., 2021; Chen et al., 2022; Ho et al., 2022b; Zhou et al., 2021).

The core idea of diffusion-based generative models is to start from a pure Gaussian noise image, and gradually modify it using a denoising network and a controlled random perturbation until it is finally crystallized into a realistic high-quality image. While different realizations of this idea exist, we follow the notation established in (Dhariwal & Nichol, 2021). Specifically, diffusion models aim to sample from a probability distribution $p_\theta(\mathbf{x})$ that approximates a data probability $q(\mathbf{x})$ representing a given dataset. Sampling starts from a pure Gaussian noise vector $\mathbf{x}_T$, and gradually updates it into samples $\mathbf{x}_{T-1}, \mathbf{x}_{T-2}, \ldots, \mathbf{x}_2, \mathbf{x}_1$ until the final output image $\mathbf{x}_0$. Each timestep $t$ represents a fixed noise level in the corresponding image $\mathbf{x}_t$, which is a mixture of $\mathbf{x}_0$ and a white Gaussian noise vector $\boldsymbol{\epsilon}_t$, specifically

$$\mathbf{x}_t = \sqrt{\alpha_t}\mathbf{x}_0 + \sqrt{1 - \alpha_t}\boldsymbol{\epsilon}_t, \tag{3}$$

with predefined signal and noise levels $\alpha_t$ and $1 - \alpha_t$, respectively ($0 = \alpha_T < \alpha_{T-1} < \cdots < \alpha_1 < \alpha_0 = 1$). A denoising model $\boldsymbol{\epsilon}_\theta(\mathbf{x}_t, t)$ is trained to approximate $\boldsymbol{\epsilon}_t$, and is subsequently used at sampling time to model the distribution $p_\theta(\mathbf{x}_{t-1}|\mathbf{x}_t) = \mathcal{N}(\boldsymbol{\mu}_t, \sigma_t^2\mathbf{I})$, with

$$\boldsymbol{\mu}_t = \sqrt{\frac{\alpha_{t-1}}{\alpha_t}}\left(\mathbf{x}_t - \frac{1 - \frac{\alpha_t}{\alpha_{t-1}}}{\sqrt{1 - \alpha_t}}\boldsymbol{\epsilon}_\theta(\mathbf{x}_t, t)\right), \tag{4}$$

and $\sigma_t^2$ is either set to a constant value representing a bound on the possible variance of the underlying distribution (Ho et al., 2020), or learned by a neural network (Nichol & Dhariwal, 2021). This distribution enables the iterative sampling, starting from pure noise $\mathbf{x}_T$ and ending with a final image $\mathbf{x}_0$.

## 3 Motivation

### 3.1 Class-Conditional Diffusion Synthesis

We are interested in generating an image from a certain user-requested class of images, labeled $y$. Previous work in this area suggested conditioning a denoising diffusion model on an input class label, thereby obtaining $\boldsymbol{\epsilon}_\theta(\mathbf{x}_t, t, y)$, and this way conditioning the sampling sequence on the desired class label (Ho et al., 2022a). In addition, building on ideas from (Sohl-Dickstein et al., 2015; Song et al., 2021), it was suggested by (Dhariwal & Nichol, 2021) to guide the diffusion process using gradients from a classifier. Assuming access to a time-dependent (actually, noise-level-dependent) classification model that outputs $\log p_\phi(y|\mathbf{x}_t, t)$, this *classifier guidance* technique suggests incorporating the model's gradient $\nabla_{\mathbf{x}_t} \log p_\phi(y|\mathbf{x}_t, t)$ into the diffusion process. This encourages the sampling output $\mathbf{x}_0$ to be recognized as the target class $y$ by the classifier model utilized. These gradients can be further scaled by a factor $s$, corresponding to a modified distribution proportional to $p_\phi(y|\mathbf{x}_t, t)^s$. Increasing $s$ results in a sharper distribution, thereby trading off diversity for fidelity. A time-dependent classifier $\log p_\phi(y|\mathbf{x}_t, t)$ is trained for this purpose using the cross-entropy loss on noisy intermediate images $x_t$, obtained by sampling images $\mathbf{x}$ from a dataset and randomly setting $t$, controlling the noise level. The classifier is then used in conjunction with a conditional diffusion model for sampling.

### 3.2 Vanilla Classifier Guidance Shortcomings

The use of gradients of the assumed underlying data distribution $\nabla_{\mathbf{x}_t} \log q(y|\mathbf{x}_t, t)$ in the diffusion process is well-motivated by Dhariwal & Nichol (2021). However, it is unclear whether the aforementioned "vanilla" training method of a classifier encourages its gradients' proximity to those of the data distribution. In fact, it was proven by (Srinivas & Fleuret, 2020) that these model gradients can be arbitrarily manipulated without affecting the classifier's cross-entropy loss nor its accuracy. Crucially, this means that training is oblivious to arbitrary changes in model gradients. We provide this proof in Appendix B for completeness, and naturally extend it to cover time-dependent classifiers trained on noisy images. It was also previously suggested that the iterative use of such gradients is akin to a black-box adversarial attack on the Inception classifier used for

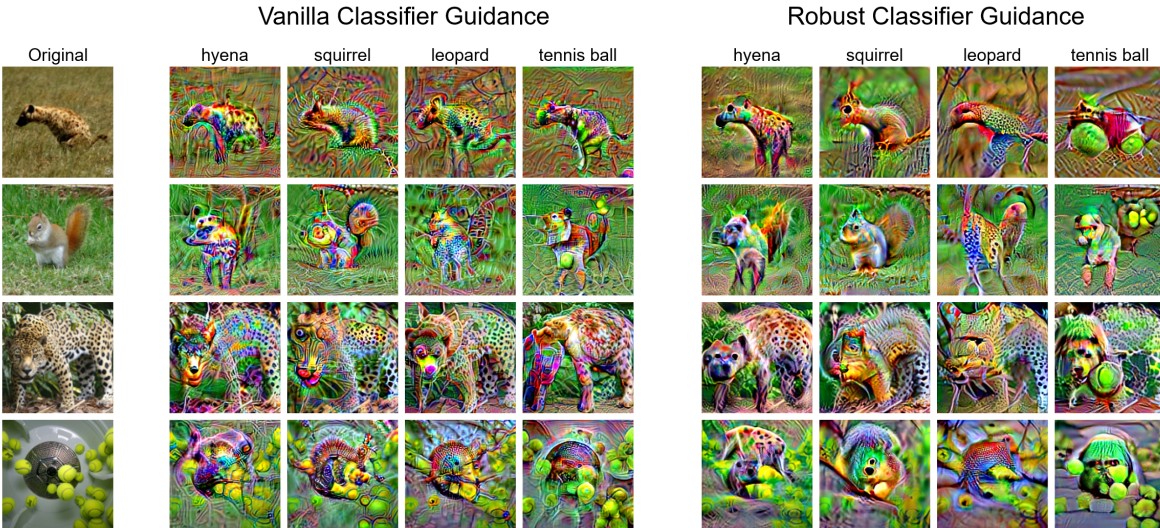

Figure 4: Maximizing the probability of target classes with given images using classifier gradients (at $t = 0$). Our robust classifier leads to images with less adversarial noise, and more aligned with the target class.

assessing generation quality (Ho & Salimans, 2021). Essentially, this may result in better nominal generation performance in metrics such as FID, while not necessarily improving the visual quality of output images. Moreover, (Chao et al., 2021) prove that changing the scaling factor of classifier guidance to $s \neq 1$ does not generally correspond to a valid probability density function.

To conclude, there is substantial evidence in recent literature towards the shortcomings of "vanilla" trained classifiers for obtaining accurate gradients. This, in turn, motivates the pursuit of alternative approximations of $\nabla_{\mathbf{x}_t} \log q(y|\mathbf{x}_t, t)$ for use as classifier guidance in diffusion-based synthesis.

## 4 Obtaining Better Gradients

### 4.1 Robust Classifier Benefits

In traditional Bayesian modeling literature, it is assumed that given a data point $\mathbf{x}$, there exists a probability of it belonging to a certain class $y$, for each possible choice of $y$. In diffusion models, the same assumption is made over noisy data points $\mathbf{x}_t$, with the probability distribution $q(y|\mathbf{x}_t, t)$. However, in practice, no concrete realization of these probabilities exists. Instead, we have access to a labeled image dataset where for each image $\mathbf{x}$, there is a "ground-truth" label $y \in \{1, \dots, C\}$. Using this data, a classifier model is encouraged to output $p_\phi(y|\mathbf{x}_t, t) = 1$ and $p_\phi(y'|\mathbf{x}_t, t) = 0$ for $y' \neq y$ through the cross-entropy loss function. While this method achieves impressive classification accuracy scores, there is no indication that its outputs would approximately match the assumed underlying distribution, nor will it reliably provide its input-gradients.

Instead of relying on "vanilla" cross-entropy training of classification models, we suggest leveraging a few recently discovered advantages of robust adversarially-trained classifiers, which have been largely unexplored in the context of diffusion models hitherto. Tsipras et al. (2019) show that traditionally trained classifiers can very easily mismatch an underlying synthetic data distribution by relying on non-robust weakly correlated features. In contrast, an adversarially-trained robust classifier would be vastly more likely to rely on more robust and highly informative features.

Interestingly, by migrating to a robust classifier, we can leverage its recently discovered phenomenon of perceptually aligned gradients (Tsipras et al., 2019). These gradients have allowed robust classifiers to be used in tasks such as inpainting, basic image generation (Santurkar et al., 2019), and boosting existing generative models (Ganz & Elad, 2021). Notably, such tasks imply the existence of decent generative

Table 1: Quality metrics for image synthesis using a class-conditional diffusion model on ImageNet ($128 \times 128$). Left to right: no guidance, vanilla classifier guidance, robust classifier guidance (ours).

| Metric | Unguided | Vanilla | Robust |
|---|---|---|---|
| Precision ($\uparrow$) | 0.70 | 0.78 | **0.82** |
| Recall ($\uparrow$) | **0.65** | 0.59 | 0.56 |
| FID ($\downarrow$) | 5.91 | 2.97 | **2.85** |

capabilities implicit in robust classifier gradients, but not "vanilla" ones. Therefore, we propose replacing the classifier used in (Dhariwal & Nichol, 2021) with a robust one.

## 4.2 Proposed Method

Note that an off-the-shelf adversarially trained robust classifier would not fit our purpose in this context. This is due to the fact that in the diffusion process, the classifier operates on intermediate images $\mathbf{x}_t$, which are a linear mixture of an ideal image and Gaussian noise. Furthermore, this mixture is also a function of $t$, which requires the classifier model to be time-dependent. Consequently, we propose the training of a novel robust time-dependent classifier model $\mathbf{h}_\phi(\mathbf{x}_t, t) = \{\log p_\phi(j|\mathbf{x}_t, t)\}_{j=1}^C$. For each sample $\mathbf{x}$ from a training set $\mathcal{D} = \{\mathbf{x}_i, y_i\}_{i=1}^N$ and timestep $t$, we first transform $\mathbf{x}$ into its noisy counterpart $\mathbf{x}_t$, and then apply a gradient-based adversarial attack on it. Since the training images are perturbed with both Gaussian and adversarial noises, we apply early stopping – the attack stops as soon as the model is fooled. Finally, the model is shown the attacked image $\tilde{\mathbf{x}}_t = \mathbf{A}(\mathbf{x}_t, \phi)$, and is trained using the cross-entropy loss with the ground-truth label $y$. The resulting loss function is formulated as

$$\mathbb{E}_{(\mathbf{x},y)\sim\mathcal{D}, t\sim\mathrm{Uni}[0,T], \mathbf{x}_t \sim q_t(\mathbf{x}_t|\mathbf{x})} \left[ \mathcal{L}_{CE}\left(\mathbf{h}_\phi\left(\tilde{\mathbf{x}}_t, t\right), y\right)\right]. \tag{5}$$

Early stopping is crucial to this training scheme, as heavily noisy images (especially for large $t$) with a full-fledged attack can easily overwhelm the model in the early stages of training. Conversely, early stopping allows the model to train on non-attacked samples initially, then proceeding to gradually more challenging cases as training progresses.

This scheme resolves several previously mentioned issues with vanilla classifier guidance. First, the gradients of an adversarially trained classifier cannot be arbitrarily manipulated like those of a vanilla model. Unlike the vanilla setting, the loss for an adversarially trained classifier is directly dependent on the adversarial attack employed, which in turn depends on the model gradients. Therefore, any change to the model's gradients, such as the one suggested by (Srinivas & Fleuret, 2020), would necessarily affect the model's predictions and loss during training. Second, gradients from a robust classifier are shown to be aligned with human perception. Namely, they exhibit salient features that humans naturally associate with the target class, and this should be contrasted with adversarial noise unintelligible by humans. Therefore, they cannot be thought of as "adversarial" to performance metrics, as their vanilla classifier counterparts (Ho & Salimans, 2021). Third, while vanilla cross-entropy-trained classifier gradients are not known to contain any intelligible features, robust classifier gradients may be interpretable. Elliott et al. (2021) have leveraged robust classifier gradients in order to highlight salient features and explain neural network decisions. These findings hint towards the superiority of these gradients.

Note that because of the need to work on intermediate images, the classifiers employed with diffusion models train on data mixed with Gaussian noise. It was discovered that this simple data augmentation can lead to gradients that are more interpretable by humans (Kaur et al., 2019), albeit to a lesser extent than observed in adversarially trained models. Therefore, we hypothesize that utilizing a model with better "perceptually aligned gradients" will yield enhanced image synthesis results.

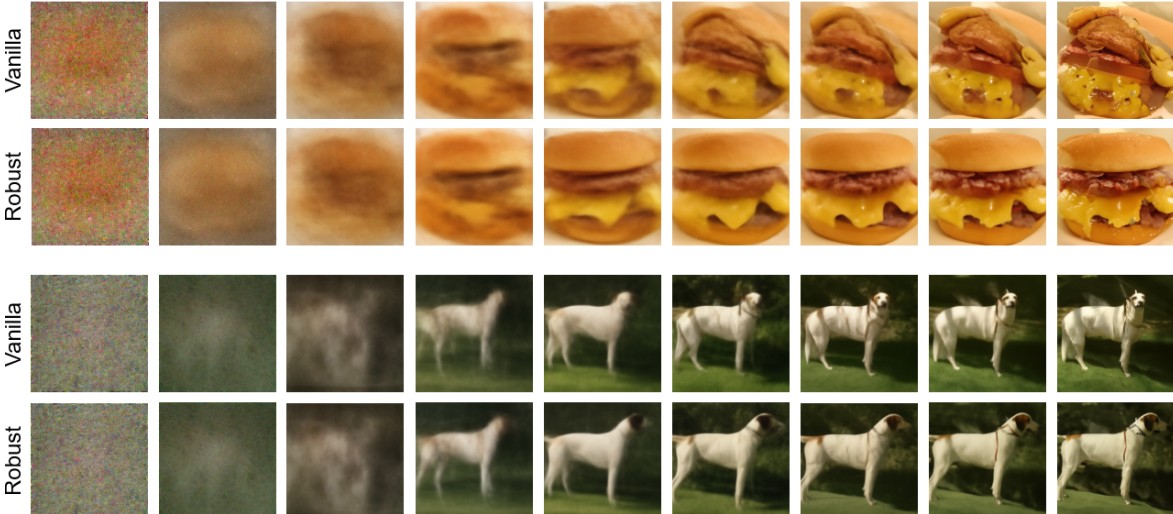

Figure 5: Approximations of the final image at uniformly spaced intermediate steps of the guided diffusion process, for the same class and the same random seed. Our robust classifier provides better guidance.

## 5 Experiments

### 5.1 Robust Time-Dependent Classifier Training

In our experiments, we focus on the highly challenging ImageNet (Deng et al., 2009) dataset for its diversity and fidelity. We consider the $128 \times 128$ pixel resolution, as it provides a sufficiently high level of details, while still being computationally efficient. In order to test our hypothesis, we require the training of a robust time-dependent classifier. We adopt the same classifier architecture from (Dhariwal & Nichol, 2021) and train it from scratch using the proposed loss in Equation (5) on the ImageNet training set. We use the gradient-based PGD attack to perturb the noisy images $\mathbf{x}_t$. The attack is restricted to the threat model $\{\mathbf{x}_t + \boldsymbol{\delta} \mid \|\boldsymbol{\delta}\|_2 \leq 0.5\}$, and performed using a step size of 0.083, and a maximum number of 7 iterations. We stop the PGD attack on samples as soon as it is successful in achieving misclassification. This early stopping technique allows the model to train on unattacked data points at the beginning, and progressively increase its robustness during training. We train the classifier for $240k$ iterations, using a batch size of 128, a weight decay of 0.05, and a linearly annealed learning rate starting with $3 \times 10^{-4}$ and ending with $6 \times 10^{-5}$. Training is performed on two NVIDIA A40 GPUs. In addition, we conduct an ablation study on CIFAR-10 (Krizhevsky et al., 2009) and report the results and implementation details in Appendix E.

To qualitatively verify that the resulting model indeed produces perceptually aligned gradients, we examine the gradients at different timesteps for a handful of natural images from the ImageNet validation set. For comparison, we also show the same gradients as produced from the vanilla classifier trained by Dhariwal & Nichol (2021). As can be seen in Figure 3, the gradients from our robust model are more successful than their vanilla counterpart at highlighting salient features aligned with the image class, and with significantly less adversarial noise. To further demonstrate the information implicit in these gradients, we perform a targeted PGD process with 7 steps and a threat model $\{\mathbf{x} + \boldsymbol{\delta} \mid \|\boldsymbol{\delta}\|_2 \leq 100\}$, maximizing a certain target class for a initial image. Figure 4 shows that our model yields images that align better with the target class.

### 5.2 Robust Classifier Guided Image Synthesis

The main goal of our work is improving class-conditional image synthesis. Following (Dhariwal & Nichol, 2021), we utilize 250 diffusion steps out of the trained 1000 (by uniformly skipping steps at sampling time) of their pre-trained conditional diffusion model for this task, while guiding it using our robust classifier. For the classifier guidance scale, we sweep across values $s \in \{0.25, 0.5, 1, 2\}$ and find that $s = 1$ produces better

Table 2: Percentage of image pairs where human evaluators prefer our robust classifier's output, the vanilla one, or have no preference. An output is considered preferred if the percentage of users who selected it passes a certain threshold.

| Threshold | Robust | Vanilla | No Preference |
|-----------|--------|---------|---------------|
| 50% | 61.5% | 31.5% | 7.0% |
| 60% | 51.0% | 28.5% | 28.5% |
| 70% | 35.0% | 13.5% | 51.5% |
| 80% | 21.5% | 8.5% | 70.0% |

results, aligning well with theoretical findings set forth by Chao et al. (2021). Dhariwal & Nichol (2021) perform a similar sweep for their model, and find that $s = 0.5$ provides the best results. In all comparisons, we use $s = 1$ for our robust classifier and $s = 0.5$ for the vanilla one. We conduct an ablation study regarding the important hyperparameters and design choices in Appendix E.

Qualitatively, the resulting synthesized images using our robust classifier look visually pleasing, as evident in Figures 1, 2, and 5. However, our method underperforms in a handful of cases, as we show in Figure 6. In order to quantify the quality of our results, we adopt the standard practice in class-conditional ImageNet image synthesis: we randomly generate 50000 images, 50 from each of the 1000 classes, and evaluate them using several well-known metrics – FID (Heusel et al., 2017), Precision, and Recall (Kynkäänniemi et al., 2019). Precision quantifies sample fidelity as the fraction of generated images that reside within the data manifold, whereas Recall quantifies the diversity of samples as the fraction of real images residing within the generated image manifold. FID provides a comprehensive metric for both fidelity and diversity, measuring the distance between two image distributions (real and generated) in the latent space of the Inception V3 (Szegedy et al., 2016) network. In Table 1, we compare three guidance methods for class-conditional generation: (i) using the class-conditional diffusion model without guidance; (ii) guidance using the pre-trained vanilla classifier; and (iii) guidance by our robust classifier. Our proposed model achieves better FID and Precision than both competing techniques, but it is outperformed in Recall.

Seeking a more conclusive evidence that our method leads to better visual quality, we conduct an opinion survey with human evaluators. We randomly sample 200 class labels, and then randomly generate corresponding 200 images using the conditional diffusion model guided by two classifiers: once using the pre-trained vanilla classifier, and once using our robust one. Both generation processes are performed using the same random seed. Human evaluators were shown a randomly ordered pair of images (one from each classifier), the requested textual class label, and the question: *"Which image is more realistic, and more aligned to the description?"*. Evaluators were asked to choose an option from "Left", "Right", and "Same". The image pairs were shown in a random order for each evaluator.

The main findings of the survey are summarized in Table 2. In each pair, we consider an image to be preferred over its counterpart if it is selected by more than a certain percentage (threshold) of users who selected a side. We then calculate the percentage of pairs where evaluators prefer our robust classifier's output, the vanilla one, or have no preference. We vary the threshold from 50% up to 80%, and observe that humans prefer our classifier's outputs over the vanilla ones for all threshold levels. During the survey, each pair was rated by 19 to 36 evaluators, with an average of 25.09 evaluators per pair, totaling 5018 individual answers. Out of these, 40.4% were in favor of our robust classifier and 32.4% were in favor of the vanilla one. These results provide evidence for human evaluators' considerable preference for our method, for a significance level of $> 95\%$.

## 6 Related Work

In this work we propose to harness the perceptually aligned gradients phenomenon by utilizing robust classifiers to guide a diffusion process. Since this phenomenon's discovery, several works have explored the generative capabilities of such classifiers. Santurkar et al. (2019) demonstrated that adversarially robust

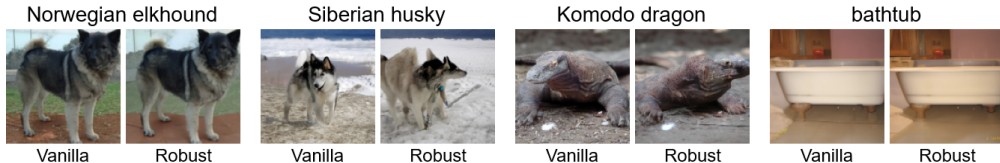

Figure 6: In a handful of cases, vanilla classifier guidance produces better outputs than robust classifier guidance. These results mostly consist of better lighting, saturation, and image focus.

models can be used for solving various generative tasks, including basic synthesis, inpainting, and super-resolution. Zhu et al. (2021) drew the connection between adversarial training and energy-based models and proposed a joint energy-adversarial training method for improving the generative capabilities of robust classifiers. Furthermore, Ganz & Elad (2021) proposed using a robust classifier for sample refinement as a post-processing step. In contrast to these works, this paper is the first to combine a robust classifier into the inner-works of a generative model's synthesis process, leading to a marked improvement.

Also worth mentioning are few other improved guidance techniques for class-conditional diffusion that were proposed. Chao et al. (2021) developed a new objective for training a classifier to capture the likelihood score. More specifically, they introduced a two-stage training scheme: first train a diffusion model, and then train the classifier to better supplement the estimations of the frozen diffusion model. Despite the intriguing idea, they demonstrate it only on low-resolution datasets ($32 \times 32$), and the two training phases are sequential, as the classifier's training requires a pre-trained diffusion model. Hence, these phases are not parallelizable. In contrast, we propose an independent classifier training scheme, which scales well to more diverse datasets with higher resolutions. Another fascinating work is (Ho & Salimans, 2021), which proposed a class-conditional synthesis without using a classifier. Instead, they offered to combine predictions of conditional and unconditional diffusion models linearly. Interestingly, a single neural network was used for both models. However, while it shows impressive performance, the proposed combination is heuristic, and requires careful hyperparameter tuning. Their work (named classifier-free guidance) follows a different research direction than ours, as we focus on enhancing classifier guidance, enabling information from outside the trained diffusion model to be incorporated into the generation process. This approach improves the generative process' modularity and flexibility, as it allows the continued definitions of more classes, without requiring further training of the base generative diffusion model. Moreover, in the case where classes are not mutually exclusive, classifier guidance allows for the generation of multiple classes in the same image at inference time (by taking the gradient for all requested classes). This is possible in classifier-free guidance only by defining a combinatorial number of class embeddings (to account for all possible class intersections).

# 7 Conclusion

In this paper we present the fusion of diffusion-based image synthesis and adversarially robust classification. Despite the success of the classifier guidance of diffusion models (Dhariwal & Nichol, 2021), we highlight several key weaknesses with this approach. Specifically, our analysis of the vanilla classifier gradients used for guidance exposes their limited ability to contribute to the synthesis process. As an alternative, we train a novel adversarially robust time-dependent classifier. We show that this scheme resolves the issues identified in vanilla classifier gradients, and use the resulting robust classifier as guidance for a generative diffusion process. This is shown to enhance the performance of the image synthesis on the highly challenging ImageNet dataset (Deng et al., 2009), as we verify using standard evaluation metrics such as FID (Heusel et al., 2017), as well as a generative performance evaluation survey, where human raters show a clear preference towards images generated by the robust classifier guidance technique that we propose.

Our future work may focus on several promising directions: (i) generalizing this technique for obtaining better gradients from multi-modal networks such as CLIP (Radford et al., 2021), which help guide text-to-image diffusion models (Ramesh et al., 2022); (ii) implementing robust classifier guidance beyond diffusion models, *e.g.* for use in classifier-guided GAN training (Sauer et al., 2022); (iii) extending our proposed technique to unlabeled datasets; and (iv) seeking better sources of perceptually aligned gradients (Ganz et al., 2022), so as to better guide the generative diffusion process.

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

# A    Additional Samples

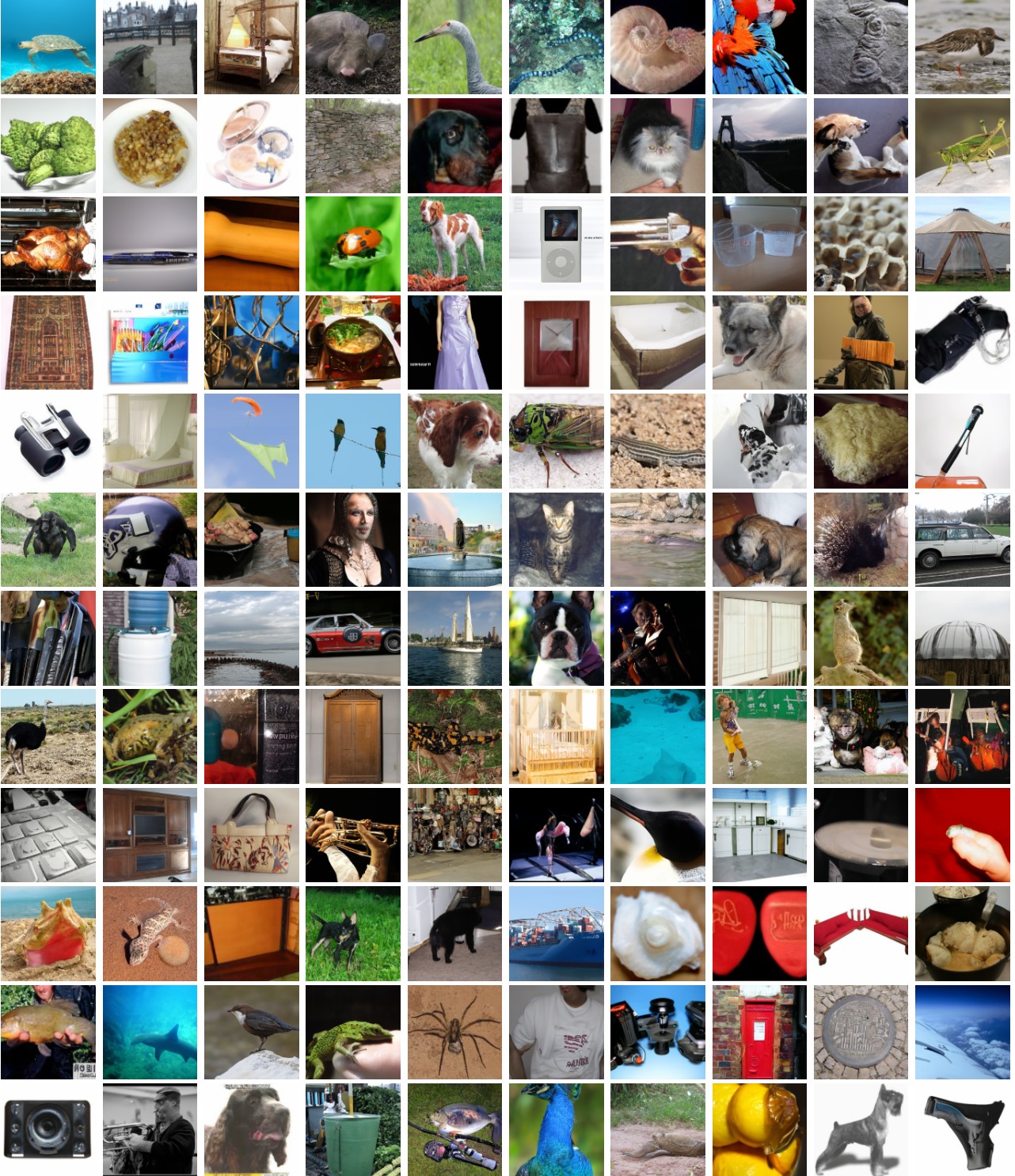

Figure 7: Uncurated samples generated by a conditional diffusion model, guided by our robust time-dependent classifier.

# B    Proof for Arbitrarily Manipulated Gradients

Below we show a theoretical observation, adapted from the original work it was presented in (Srinivas & Fleuret, 2020).

**Observation.**  Assume a neural network classifier $\mathbf{h}(\mathbf{x}) = \{h_i(\mathbf{x})\}_{i=1}^C$, where $h_i : \mathbb{R}^d \to \mathbb{R}$, and an arbitrary function $g : \mathbb{R}^d \to \mathbb{R}$. Consider another neural network classifier $\tilde{\mathbf{h}}(\mathbf{x}) = \{\tilde{h}_i(\mathbf{x})\}_{i=1}^C$, where $\tilde{h}_i(\mathbf{x}) = h_i(\mathbf{x}) + g(\mathbf{x})$, for which we obtain $\nabla_\mathbf{x}\tilde{h}_i(\mathbf{x}) = \nabla_\mathbf{x}h_i(\mathbf{x}) + \nabla_\mathbf{x}g(\mathbf{x})$. For this, the corresponding cross-entropy loss values and accuracy scores remain unchanged.

*Proof.* For any data point $\mathbf{x} \in \mathbb{R}^d$ and its corresponding true label $y \in \{1, \dots, C\}$, the cross-entropy loss for the neural network $\mathbf{h}(\mathbf{x})$ is given as

$$
\begin{aligned}
\mathcal{L}_{CE}(\mathbf{h}(\mathbf{x}), y) &= -\log \frac{\exp(h_y(\mathbf{x}))}{\sum_{j=1}^C \exp(h_j(\mathbf{x}))} \\
&= -\log \exp(h_y(\mathbf{x})) + \log \left( \sum_{j=1}^C \exp\left(h_j(\mathbf{x})\right) \right) \\
&= -h_y(\mathbf{x}) + \log \left( \sum_{j=1}^C \exp\left(h_j(\mathbf{x})\right) \right).
\end{aligned}
\tag{6}
$$

For the neural network $\tilde{\mathbf{h}}(\mathbf{x})$, we obtain the cross-entropy loss by substituting $\tilde{h}_i(\mathbf{x}) = h_i(\mathbf{x}) + g(\mathbf{x})$ in Equation (6),

$$
\begin{aligned}
\mathcal{L}_{CE}(\tilde{\mathbf{h}}(\mathbf{x}), y) &= -\tilde{h}_y(\mathbf{x}) + \log \left( \sum_{j=1}^C \exp\left(\tilde{h}_j(\mathbf{x})\right) \right) \\
&= -h_y(\mathbf{x}) - g(\mathbf{x}) + \log \left( \sum_{j=1}^C \exp\left(h_j(\mathbf{x}) + g(\mathbf{x})\right) \right) \\
&= -h_y(\mathbf{x}) - g(\mathbf{x}) + \log \left( \sum_{j=1}^C \exp\left(h_j(\mathbf{x})\right) \exp\left(g(\mathbf{x})\right) \right) \\
&= -h_y(\mathbf{x}) - g(\mathbf{x}) + \log \left( \sum_{j=1}^C \exp\left(h_j(\mathbf{x})\right) \right) + \log\left(\exp\left(g(\mathbf{x})\right)\right) \\
&= -h_y(\mathbf{x}) - g(\mathbf{x}) + \log \left( \sum_{j=1}^C \exp\left(h_j(\mathbf{x})\right) \right) + g(\mathbf{x}) \\
&= -h_y(\mathbf{x}) + \log \left( \sum_{j=1}^C \exp\left(h_j(\mathbf{x})\right) \right) \\
&= \mathcal{L}_{CE}(\mathbf{h}(\mathbf{x}), y).
\end{aligned}
\tag{7}
$$

The last equality holds due to Equation (6). It also holds that

$$
\arg\max_{i\in\{1,\dots,C\}} \tilde{h}_i(\mathbf{x}) = \arg\max_{i\in\{1,\dots,C\}} \left(h_i(\mathbf{x}) + g(\mathbf{x})\right) = \arg\max_{i\in\{1,\dots,C\}} h_i(\mathbf{x}),
\tag{8}
$$

implying identical predictions for both networks on any input, and therefore identical accuracy scores, completing the proof.

□

This observation shows that two neural networks with an identical loss over all inputs, can have arbitrarily different gradients, as the proof does not assume any limitations on $g(\mathbf{x})$ nor on $\mathbf{x}$. This also implies that the proof remains valid for noisy training data, and as a result, it remains valid for time-dependent classifiers which follow a Gaussian noise schedule, such as the one trained by (Dhariwal & Nichol, 2021). However, when transitioning into an adversarial training scheme, the perturbed training inputs become dependent on the model's gradients, and consequently, the adversarial loss presented in Equation (5) changes. This motivates the use of an adversarially-trained classifier over a vanilla one, when the goal is to obtain better gradients.

## C   Opinion Survey Details

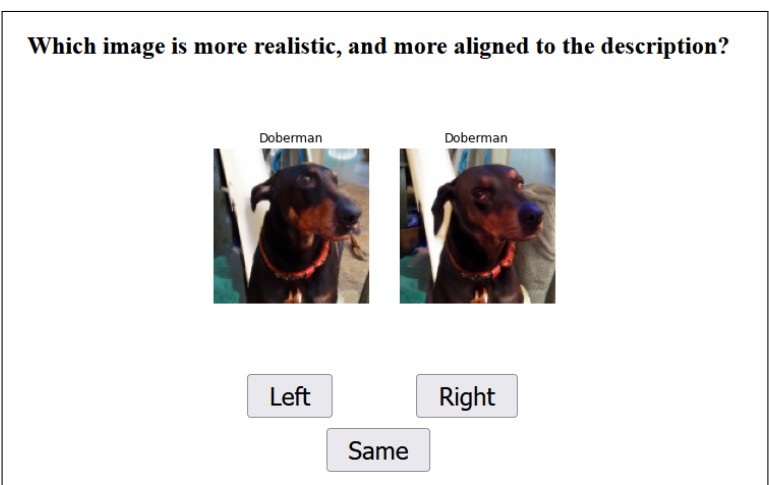

Figure 8: A screenshot of the survey that was shown to human evaluators.

As previously mentioned, we conducted an opinion survey, asking human evaluators to choose their preferred image out of a pair of generated images or to choose that they have no preference. Evaluators were shown 200 pairs, randomly shuffled, where each pair consists of two randomly ordered images generated with the same class label and seed, once guided by our robust classifier, and once guided using the pre-trained vanilla classifier. An example of the screen shown to evaluators is displayed in Figure 8.

Some evaluators did not fill out the entire survey. Hence, different pairs have different numbers of answers. Nevertheless, sufficient data was collected for all pairs, as the number of answers per pair ranged from 19 to 36, averaging 25.09 answers.

## D   Implementation Details

We base our implementation on the publicly available code provided by (Dhariwal & Nichol, 2021). For implementing the adversarial training scheme, we adapt code from (Madry et al., 2018) to work with time-dependent models and enable early stopping, and use it to train our robust time-dependent classifier. The sampling routine is identical to that of (Dhariwal & Nichol, 2021), up to changing the model and sampling hyperparameters. Our code and trained robust time-dependent classifier models are available at `https://github.com/bahjat-kawar/enhancing-diffusion-robust`.

## E   Ablation Study

To perform a comprehensive ablation study on the different hyperparameters of our method, we consider the smaller CIFAR-10 (Krizhevsky et al., 2009) dataset, containing $32 \times 32$-pixel images. We train a class-

Table 3: FID at different training iterations of an unguided class-conditional diffusion model for CIFAR-10.

| Iterations | 50k | 100k | 150k | 200k | 250k | 300k |
|---|---|---|---|---|---|---|
| **FID** | 11.65 | 8.41 | 7.81 | 7.67 | 7.68 | 7.91 |

conditional diffusion model for CIFAR-10, with an architecture adapted from Nichol & Dhariwal (2021) (changing the dropout from 0.3 to 0.1, making the model class-conditional, and using a linear 1000-step noise schedule akin to Dhariwal & Nichol (2021)). We measure FID with 10000 images against the validation set, and show the diffusion model's results in Table 3. The trained diffusion model achieves its best FID at 200k training iterations. As a general point of reference, StyleGAN2 (Karras et al., 2020) achieves an FID of 11.07 (Zhao et al., 2020). As a baseline, we train a vanilla time-dependent classifier with an architecture similar to Dhariwal & Nichol (2021) (changing: the number of output channels to 10 and image size to 32 to match CIFAR-10, the attention resolutions from $[32, 16, 8]$ to $[16, 8]$, and the classifier width from 128 to 32). At 200k iterations and $s = 0.25$, it achieves an FID of 7.65 with the aforementioned diffusion model.

Table 4: FID on CIFAR-10 for different threat models and numbers of attacker steps, after training a robust classifier for 200k steps and using a guidance scale of $s = 0.25$. Best result is highlighted in **bold**.

| Threat Model | 5 steps | 7 steps | 9 steps |
|---|---|---|---|
| $\{\mathbf{x}_t + \boldsymbol{\delta} \mid \|\boldsymbol{\delta}\|_2 \le 0.25\}$ | 7.62 | 7.62 | 7.64 |
| $\{\mathbf{x}_t + \boldsymbol{\delta} \mid \|\boldsymbol{\delta}\|_2 \le 0.5\}$ | 7.62 | **7.60** | 7.63 |
| $\{\mathbf{x}_t + \boldsymbol{\delta} \mid \|\boldsymbol{\delta}\|_2 \le 1.0\}$ | 7.64 | 7.63 | 7.65 |
| $\{\mathbf{x}_t + \boldsymbol{\delta} \mid \|\boldsymbol{\delta}\|_\infty \le 4/255\}$ | 7.61 | 7.61 | 7.62 |
| $\{\mathbf{x}_t + \boldsymbol{\delta} \mid \|\boldsymbol{\delta}\|_\infty \le 8/255\}$ | 7.65 | 7.64 | 7.67 |

Then, we adversarially train a robust classifier with the same architecture. We perform an ablation study on the different adversarial training hyperparameters and summarize our results in Table 4. The attack step size is set to 2.5 times the upper bound in the attack threat model, divided by the number of attacker steps. We choose the best performing robust classifier, which was trained on the threat model $\{\mathbf{x}_t + \boldsymbol{\delta} \mid \|\boldsymbol{\delta}\|_2 \le 0.5\}$ with 7 attacker steps and 200k training iterations.

Table 5: FID for classifier guidance scales ($s$) for both the vanilla and robust classifiers on CIFAR-10.

| Classifier | $s = 0$ | $s = 0.0625$ | $s = 0.125$ | $s = 0.25$ | $s = 0.5$ |
|---|---|---|---|---|---|
| Vanilla | 7.67 | 7.65 | 7.62 | 7.65 | 7.87 |
| Robust | 7.67 | 7.62 | 7.58 | 7.60 | 7.79 |

Moreover, we also conduct an ablation study for the classifier guidance scale hyperparameter $s$, for both the vanilla and robust classifiers, and present the results in Table 5. Our robust classifier outperforms its vanilla counterpart in every classifier scale, attaining an FID of 7.58 at $s = 0.125$. These results show that a simple traversal of the classifier scales cannot improve the vanilla classifier's performance to the level attained by the robust one.

## E.1 Effect of Classifier Guidance Scale on Precision and Recall

We measure the effect of the classifier guidance scale hyperparameter $s$ on the Precision and Recall metrics for our main ImageNet experiments for both the vanilla and robust classifiers. The results are presented in Table 6.

Table 6: Precision and Recall for classifier guidance scales ($s$) for both the vanilla and robust classifiers on ImageNet.

| Classifier | Metric | $s = 0$ | $s = 0.25$ | $s = 0.5$ | $s = 1.0$ | $s = 2.0$ | $s = 4.0$ |
|---|---|---|---|---|---|---|---|
| Vanilla | Precision | 0.70 | 0.73 | 0.77 | 0.80 | 0.84 | 0.86 |
| | Recall | 0.65 | 0.62 | 0.59 | 0.53 | 0.46 | 0.37 |
| Robust | Precision | 0.70 | 0.74 | 0.77 | 0.82 | 0.86 | 0.89 |
| | Recall | 0.65 | 0.62 | 0.60 | 0.56 | 0.47 | 0.39 |

Diffusion models are known for their great mode coverage, which gives them an edge in the Recall metric. In fact, the unguided diffusion model achieves better Recall rates than any guided version. As $s$ increases (in both vanilla and robust settings), we trade off Recall for better Precision.

Our robust model at $s = 0.5$ achieves Precision and Recall that are similar to Dhariwal & Nichol (2021). However, in FID, which computes a distance between distributions (covering concepts from both Precision and Recall), our robust classifier (at $s = 1.0$) improves upon the vanilla one (which achieves its best FID result at $s - 0.5$), meaning that the improvement in precision outweighs the loss in recall.

