# OpenReview forum: "Enhancing Diffusion-Based Image Synthesis with Robust Classifier Guidance"
_TMLR — Accepted by TMLR_

### Review · Reviewer_s4sm · 2022-12-09

**Summary Of Contributions:**

This work addresses a shortcoming of diffusion-based image synthesis which rely on an image classification model for (class) guided image generation. The authors argue that well known adversarial vulnerability of such image classification models also impairs the performance of image synthesis approaches based on a vulnerable model. The authors propose to utilize adversarial hardened (here in form of adversarial training) classification models instead. The authors provide some quantitative as well as qualitative evidence to show improvements of their method for image synthesis.

**Audience:**

Yes

**Broader Impact Concerns:**

Similar to other image generation methods the proposed method can be used to also improve quality of faked images / photos and can hence be misused.

**Claims And Evidence:**

No

**Requested Changes:**

I consider it critical that some further quantitative evidence / ablations are provided to better understand the achieved improvements for the new approach.

If addressed I would raise my "Claims And Evidence" rating "No" -> "Yes"



**Strengths And Weaknesses:**

Strengths:

• The paper addresses an important short-coming of guided image-synthesis approaches and provides a solution to address these short-comings as well as a valid line of argument why / how their solution addresses the short-coming. To the best of my knowledge the proposed solution is novel wrt state-of-the-art approaches.

• The paper provides some strong qualitative evidence that their proposed approach improves guided image synthesis.

• Implementation of the approach and reproduction of the findings seem straight forward, however no code is provided.

• The paper is well written and structured. The main contributions of the paper are introduced and presented in an easy to access manner.


Weaknesses:

• My major concern for this work is the lack of sufficient quantitative results / evaluations. Only Table 1 provides some rather limited quantitative evaluation. Specifically additional ablations studies / argumentation would be desirable addressing the low recall rate for the proposed approach. Like why/how and in which cases this doesn’t work.


Minor Weakness:

• I find the term “time-dependent” rather confusing. I understand that this refers to the iterations in the diffusion model, but is suggests a relation to video-generation, which is not the meaning here. If “time-dependent” is a commonly used term in the context of diffusion models that is fine.

---

> ### Author Response · Authors · 2022-12-20
> **Author reply to Reviewer s4sm**
>
> Thank you for your valuable input. We are pleased that you highlighted the key strengths of our paper, and hope to improve the parts which you suggested improving.
>
> **Code**: We will release code upon acceptance, along with our trained robust classifier and 50,000 images for FID evaluation. Note that our code is similar to (Dhariwal & Nichol, 2021), so it is straightforward to implement as you have mentioned.
>
> **"time-dependent"**: The term "time-dependent" indeed refers to iterations of the diffusion model, and it is in use in the broader diffusion model literature. We will edit our paper to better introduce this term to future readers.
>
> **More evidence**: We would be happy to present more ablations to support our claims. In the paper, we provide quantitative results in the form of FID, Precision, and Recall in Table 1, as well as the human evaluation study in Table 2. We believe the human evaluation study to be equally important, as the main goal of image generation is to synthesize images that look real to humans. Our study shows that humans prefer our method's outputs over the baseline's at all thresholds. This result, as well as the improved FID, attest to the improvement achieved by our approach.
>
> To address the recall rate, we generate 50,000 images and examine the FID at varying classifier guidance scales ($s$), and obtain the following results:
>
> | Scale ($s$) | Prec | Recall |
> |-|-|-|
> | 0 (Unguided) | 0.70 | 0.65 |
> 0.25	|	0.74	|	0.62 |
> 0.50	|	0.77	|	0.60 |
> 1.00	|	0.82	|	0.56 |
> 2.00	|	0.86	|	0.47 |
> 4.00	|	0.89	|	0.39 |
>
> Diffusion models are known for their great mode coverage, which gives them an edge in the recall metric. In fact, the unguided diffusion model achieves better recall rates than any guided version. When we use $s=0.5$, the same as in (Dhariwal & Nichol, 2021), we achieve similar results to them in precision and recall. However, in FID, which computes a distance between distributions (covering concepts from both precision and recall), our robust classifier (at $s=1.0$) improves upon the vanilla one (which achieves its best FID result at $s=0.5$), meaning that the improvement in precision outweighs the loss in recall.
>
> If there are additional ablations that you had in mind that could further support the paper, beyond the evidence provided by FID, Precision, and human evaluation, please let us know and we would be happy to perform the necessary experiments for them.

---

> > ### Comment · Reviewer_s4sm · 2022-12-21
> > **Comment on ablation**
> >
> > I thank the authors for addressing my concerns.
> >
> > Additional remarks on the ablations:
> >
> > * I would suggest to add the new table (varying s for the robust version) also for the vanilla setting and replace the existing table in the main manuscript.
> >
> > * Would it be possible to perform an ablation study on the impact of different adversarial attack settings (different attackers, etc.)?
> >
> > * As an additional qualitative figure, could you add some failure cases for your method?

---

> > > ### Author Response · Authors · 2023-01-02
> > > **Author response**
> > >
> > > Thank you for the clarification.
> > >
> > > **Varying $s$:** We have generated the same table for the vanilla classifier as well, as per your request:
> > > | Scale ($s$) | Prec | Recall |
> > > |-|-|-|
> > > | 0 (Unguided) | 0.70 | 0.65 |
> > > 0.25    |    0.73    |    0.62 |
> > > 0.50    |    0.77    |    0.59 |
> > > 1.00    |    0.80    |    0.53 |
> > > 2.00    |    0.84    |    0.46 |
> > > 4.00    |    0.86    |    0.37 |
> > > While the classifier scale is able to trade off precision for recall in both settings, the robust classifier achieves a better tradeoff and thus a better FID (which describes the overall closeness of the generated image distribution to the test set). We will add both tables to the paper in a revision.
> > >
> > > **Attacker ablation:** Thank you for your suggestion. We have started training classifiers adversarially under different settings (parameters, attack strategies). Such a training takes time, and we are unable to report results in the span of the 2-week rebuttal period. However, we will add such an ablation to a revision of the paper.
> > >
> > > **Failure cases:** Indeed, we can. We examine several images from the survey (available in the supplementary material) where users preferred the vanilla version:
> > > - Pair 10 (vanilla is right)
> > > - Pair 11 (vanilla is left)
> > > - Pair 23 (vanilla is left)
> > > - Pair 38 (vanilla is left)
> > > - Pair 33 (vanilla is right)
> > > - Pair 142 (vanilla is right)
> > >
> > > Note that in most of these (e.g., 10, 11, 44), the vanilla generation outperformed the robust one in terms of overall image focus, saturation, and lighting, while the image contents were very similar in both results.
> > > We will aggregate these into a figure and a brief discussion in a revision.

---

> ### Author Response · Authors · 2023-01-17
> **Additional results**
>
> Dear reviewer,
>
> We have added more results in the OpenReview comment titled "Additional datasets and ablations". We hope the additional results address the remaining concerns.
>
> Best regards,
>
> Authors.

---

### Review · Reviewer_KbY7 · 2022-12-22

**Summary Of Contributions:**

This paper proposes a methodology to improve image quality in class-conditional DDPMs. In particular, they rely on the observation from prior work that adversarially robust classifiers tend to have more perceptually-aligned gradients. They leverage this property by using gradients from a robust classifier (rather than a standard ERM-trained classifier) to guide the diffusion process (based on the setup of Dhariwal and Nichol). They validate their approach on ImageNet generation in terms of automated metrics as well as a human study.

**Audience:**

Yes

**Claims And Evidence:**

Yes

**Requested Changes:**

My main concerns are described in the three weaknesses above. I would be happy to recommend acceptance if the authors addressed those concerns.

**Strengths And Weaknesses:**

Strengths:

- The paper is well-written and easy to follow.
- It is interesting to see ideas from the adversarial robustness literature being employed to improve diffusion models.

Weaknesses:
- A somewhat salient point that is not clear is whether they train the diffusion model from scratch or fine-tune the pre-trained model from Dhariwal and Nichol on ImageNet. If it is the latter, (i) the authors should explain why they made this choice and (ii) it should be clearly stated throughout the paper. It would also be valuable to see how the approaches compare on training the model from scratch.
- It is not clear how significant the improvement in generation actually is. In particular, one concern I have is that the proposed method is trading of quality (improved precision and FID) for diversity (recall)---perhaps similar to the effect of the scaling factor s. Moreover, the human experiment is also setup to measure quality and not diversity. I wonder whether simply tuning "s" for the vanilla method would yield similar results.
- Finally, given that robust training is a key component of the proposed approach, the authors could do a better job of examining how choices of hyperparameters therein affect generation quality. Currently, parameters such as eps, step size, number of steps are merely stated without explanation. The paper could be strengthened with an analysis of how image quality changes with eps and step size. Also, it seems like the step size being used for the attack might be too small---in general, the rule of thumb is 2.5*eps/steps.

Some additional questions/comments:

- In Table 1, does no guidance just mean using the pre-trained model from Dhariwal and Nichol (trained with vanilla classifier guidance)?
- It seems from Table 1 that like a big part of the improvement stems from training the model longer with any guidance.
- Even though the metrics used in the paper (FID, precision and recall) are derived from prior work, it would be nice if the authors could briefly describe them at least in the Appendix.

---

> ### Author Response · Authors · 2023-01-02
> **Author reply to Reviewer KbY7**
>
> Thank you for your valuable comments. Below we address the raised concerns:
>
> **Training from scratch**: We apologize for the confusion. We trained the robust classifier model from scratch, independent of the model training of Dhariwal and Nichol. Both our robust model and Dhariwal and Nichol’s vanilla one used similar training hyperparameters (allowing for a fair comparison).
> We will revise the paper and better clarify this point in the final version.
>
> **Tuning $s$:** Indeed, precision and the human opinion survey measure image quality, whereas recall measures diversity. FID measures a distance between distributions, thereby accounting for both quality and diversity.
>
> We conducted the experiment that you suggested (for the vanilla classifier), and obtained the following results:
> | Scale ($s$) | Prec | Recall |
> |-|-|-|
> | 0 (Unguided) | 0.70 | 0.65 |
> 0.25    |    0.73    |    0.62 |
> 0.50    |    0.77    |    0.59 |
> 1.00    |    0.80    |    0.53 |
> 2.00    |    0.84    |    0.46 |
> 4.00    |    0.86    |    0.37 |
> Note that while tuning the classifier scale $s$ can trade off recall for precision, the obtained tradeoff fails to achieve a smaller distribution mismatch (FID score). We hope this experiment further consolidates the improvement incurred by using a robust classifier, which achieves a better tradeoff, resulting in a better FID score.
>
> **Attacker ablation:** Thank you for this suggestion. We have started training classifiers adversarially under different settings (parameters: eps, step size, number of steps, and attack strategies). Such a training takes time, and we are unable to report results in the span of the 2-week rebuttal period. However, we will add such an ablation to a revision of the paper.
>
> **No guidance:** No guidance means generating using the diffusion model, without the use of any classifier (neither the robust one nor the vanilla one from Dhariwal and Nichol).
>
> **Training the model longer:** Our model was trained from scratch, and it did not continue from Dhariwal and Nichol’s checkpoint.
>
> **Describing the metrics:** Thanks for the suggestion. We will better describe them in a revised version of the paper.

---

> ### Author Response · Authors · 2023-01-17
> **Additional results**
>
> Dear reviewer,
>
> We have added more results in the OpenReview comment titled "Additional datasets and ablations". We hope the additional results address the remaining concerns.
>
> Best regards,
>
> Authors.

---

### Review · Reviewer_RJzT · 2022-12-23

**Summary Of Contributions:**

The paper proposes to replace the classifier for classifier-guided diffusion models with an adversarially trained classifier, arguing that a classifier trained with adversarial noise provides better gradients and information to a class-conditional diffusion net at sample time. For this, the paper compares results of a class-conditional diffusion net on ImageNet guided by either a vanilla or an adversarially trained classifier. Using an adversarially robust classifier for guidance improves FID and precision and a human user study indicates that humans prefer images generated with the guidance of the adversarially robust classifier.

**Audience:**

No

**Claims And Evidence:**

No

**Requested Changes:**

The approach is only evaluated on a single dataset. While ImageNet is a good benchmark the approach should be evaluated on other datasets, too. The approach is somewhat constrained in that it only works on class-conditional datasets, nevertheless, there are other datastes on which this approach can be evaluated such as CUB-200, Oxfod 102-Flower, possible CelebA, or even CIFAR.

Overall, however, I believe the biggest drawback is the limitation of classifier-guided models to class-conditional image datasets in the first place. Arguably, most of the popularity of diffusion days nowadays stems from the fact that they are able to model large, diverse datasets. In these cases, classifier-guidance is not realistic since there are no available class labels for most images. Approaches such as CLIP-guidance may alleviate this to a degree but are not evaluated in this paper.

The default approach is classifier-free guidance for almost all of these cases, which has the advantage of not needing an additional classifier network and is not limited to class-conditional guidance.

As a summary, I would expect the paper to have
- comparisons on more datasets
- extension of the approach to CLIP-guidance or similar guidance mechanisms that are not restricted to class-conditional models
- a comparison to classifier-free guidance: if the approach is not better than classifier-free guidance despite being more complex and expensive I don't see the benefit

**Strengths And Weaknesses:**

The motivation of the approach is well written and makes it clear why an adversarially robust classifier may be preferred over a vanilla classifier for guidance of class-conditional diffusion models.
The evaluation is done on ImageNet, arguably one of the most challenging class-conditional datasets for image generation and based on FID and a human user study the proposed methodology seems to improve upon the baseline.

---

> ### Author Response · Authors · 2023-01-02
> **Author reply to Reviewer RJzT**
>
> Thank you for your valuable comments. Below we address the raised issues:
>
> **More datasets:** As you have mentioned, ImageNet is considered the main benchmark for image generation. Nevertheless, as per your suggestion, we will evaluate on some of the datasets you suggested to better supplement our paper. Please note that most of these datasets do not have established baselines, meaning that we need to train both diffusion models and classifiers for them. Thus, we will not be able to report results in the span of the 2-week rebuttal period, and will only add them in a later revision.
>
> **Limitation to class-conditional datasets:** Class-conditional generation has been the standard benchmark task for many years, and it serves important applications. While text-to-image generation has risen to popularity in recent months, we argue that this should not dismiss the entirety of the effort made in class-conditional generation. Moreover, our approach can be trivially extended to text-conditional generation with a CLIP model with perceptually aligned gradients. However, the development of such a model requires substantial research efforts and computational power, and we consider it out of scope for this paper.
>
> **Comparison to classifier-free guidance:** Classifier-free guidance is indeed a viable alternative to classifier guidance. However, both methods have only been suggested less than 2 years ago, and both are still active areas of research. Our approach presents an enhancement over the current leading approach for classifier guidance.
>
> While classifier-free guidance indeed requires less model parameters, classifier guidance has several advantages of its own: Classifier guidance is more modular, as it allows the continued definitions of more classes, without requiring further training of the base generative diffusion model. Moreover, in the case where classes are not mutually exclusive, it allows for the generation of multiple classes in the same image at inference time (by taking the gradient for all requested classes). This is not possible in classifier-free guidance without a significant change in its training and a combinatorial number of class embeddings (to account for all possible class intersections).
>
> We will expand upon this discussion in the final revision of our paper.

---

> ### Author Response · Authors · 2023-01-17
> **Additional results**
>
> Dear reviewer,
>
> We have added more results in the OpenReview comment titled "Additional datasets and ablations". We hope the additional results address the remaining concerns.
>
> Best regards,
>
> Authors.

---

### Author Response · Authors · 2023-01-17
**Additional datasets and ablations**

We would like to thank the reviewers for their constructive comments, as well as the action editor for allowing us more time to provide the necessary experiments.

In order to address **Reviewer RJzT**’s concern regarding additional datasets, we train the following models:
1. A class-conditional diffusion model for CIFAR-10, with an architecture similar to [1].
2. A vanilla classifier for CIFAR-10, with an architecture similar to [2].
3. A robust classifier for CIFAR-10, with the same architecture as the vanilla one.

We will note the exact architecture specifications in the appendix upon acceptance, as well as releasing the source code and trained models. For the main evaluation metric, we consider FID with 10k images against the CIFAR validation set. As a general point of reference, StyleGAN2 [3] achieves an FID of 11.07 [4].

We choose the training iteration for the diffusion model with the best performance (200k iterations) among the following:
| Iterations | FID |
|-|-|
| 50k | 11.65 |
100k    |    8.41    |
150k   |    7.81    |
200k    |    7.67    |
250k   |    7.68    |
300k    |    7.91    |
400k    |    8.65    |

Coupled with this diffusion model, we sample images using classifier guidance (with $s=0.25$) with a vanilla classifier as a baseline for our method. We choose the best performing one (200k iterations):
| Iterations | $s=0$ | $s=0.25$ |
|-|-|-|
| 150k | 7.67 | 7.67 |
200k | 7.67 | 7.65 |
250k | 7.67 | 7.66 |

Then, in order to train a robust classifier, we follow the advice of **Reviewers KbY7 & s4sm** and perform an ablation study. In order to study the training hyperparameters, we freeze the classifier scale at 0.25, and vary the following hyperparameters: (i) the attack strategy, given by the threat model (norm and $\epsilon$); and (ii) the number of steps for the attacker. The step size for the attacker is given by the rule of thumb suggested by **Reviewer KbY7** -- 2.5*eps/steps. We get the following FID results for 200k training iterations:
| Threat model | 5 steps | 7 steps | 9 steps|
|-|-|-|-|
| $L_2$, $\epsilon=0.25$ | 7.62 | 7.62 | 7.64 |
$L_2$, $\epsilon=0.5$ | 7.62 | 7.60 | 7.63 |
$L_2$, $\epsilon=1.0$ | 7.64 | 7.63 | 7.65 |
$L_\infty$, $\epsilon=4/255$ | 7.61 | 7.61 | 7.62 |
$L_\infty$, $\epsilon=8/255$ | 7.65 | 7.64 | 7.67 |

Furthermore, we try varying the number of training iterations for the best performing model ($L_2$, $\epsilon=0.5$, 7 steps):
| Iterations | FID |
|-|-|
| 150k | 7.61 |
200k    |    7.60    |
250k    |    7.62    |

Therefore, we choose $L_2$, $\epsilon=0.5$, 7 steps, 200k iterations as the robust classifier model. Finally, we conduct an ablation study for the classifier scale ($s$) on both the vanilla and robust classifiers, obtaining the following results:
| Model | $s=0$ | $s=0.0625$ | $s=0.125$ | $s=0.25$ | $s=0.5$ |
|-|-|-|-|-|-|
| Vanilla | 7.67 | 7.65 | 7.62 | 7.65 | 7.87 |
| Robust | 7.67 | 7.62 | 7.58 | 7.60 | 7.79 |

Our robust classifier outperforms its vanilla counterpart in every classifier scale, attaining an FID of 7.58 at $s=0.125$. A simple traversal of the classifier scales cannot improve the vanilla classifier’s performance to the level attained by the robust one.

Note that while the improvements here are relatively small, they are still consistent across different classifier scales. We will add this ablation study and CIFAR-10 results to our manuscript upon acceptance, in addition to publishing the code and the trained model checkpoints. We hope these experiments address any remaining concerns raised by the reviewers.


[1] Nichol, Alexander, and Prafulla Dhariwal. "Improved denoising diffusion probabilistic models." International Conference on Machine Learning. PMLR, 2021.

[2] Dhariwal, Prafulla, and Nichol, Alexander. "Diffusion models beat gans on image synthesis." Advances in Neural Information Processing Systems 34 (2021): 8780-8794.

[3] Karras, Tero, et al. "Analyzing and improving the image quality of stylegan." Proceedings of the IEEE/CVF conference on computer vision and pattern recognition. 2020.

[4] Zhao, Shengyu, et al. "Differentiable augmentation for data-efficient gan training." Advances in Neural Information Processing Systems 33 (2020): 7559-7570.

---

### Decision · Action_Editors · 2023-03-09

**Recommendation:** Accept with minor revision

**Comment:**

Reasoning behind my recommendation can be found under "Claims and Evidence"

**Audience:**

The topic of the paper is highly relevant to the TMLR community

**Claims And Evidence:**

Reviewers raised several concerns regarding insufficient empirical evidence in support of the paper's claims.  Most of the concerns seem to have been addressed by the authors' clarifications and additional experiments.  A notable unresolved point is the argument of reviewer RJzT by which necessary comparisons to classifier-free guidance models are missing.  While I agree with the reviewer that such comparisons can significantly strengthen the paper, I generally accept the authors' position by which classifier-guidance and classifier-free guidance models form two separate categories.  I do however urge the authors to thoroughly discuss this point while clearly highlighting the benefits of classifier-guidance over its classifier-free counterpart.